# Redesigning Medication Management in the Emergency Department: The Impact of Partnered Pharmacist Medication Charting on the Time to Administer Pre-Admission Time-Critical Medicines, Medication Order Completeness, and Venous Thromboembolism Risk Assessment

**DOI:** 10.3390/pharmacy12020071

**Published:** 2024-04-17

**Authors:** Tesfay Mehari Atey, Gregory M. Peterson, Mohammed S. Salahudeen, Tom Simpson, Camille M. Boland, Ed Anderson, Barbara C. Wimmer

**Affiliations:** 1School of Pharmacy and Pharmacology, College of Health and Medicine, University of Tasmania, Hobart 7005, Australia; 2Pharmacy Department, Royal Hobart Hospital, Tasmanian Health Service, Hobart 7000, Australia; 3Marinova Pty Ltd., Cambridge 7170, Australia

**Keywords:** partnered pharmacist, interdisciplinary, co-charting, medication charting, emergency department, time-critical medicine, medication order, venous thromboembolism

## Abstract

In order to enhance interdisciplinary collaboration and promote better medication management, a partnered pharmacist medication charting (PPMC) model was piloted in the emergency department (ED) of an Australian referral hospital. The primary objective of this study was to evaluate the impact of PPMC on the timeliness of time-critical medicines (TCMs), completeness of medication orders, and assessment of venous thromboembolism (VTE) risk. This concurrent controlled retrospective pragmatic trial involved individuals aged 18 years and older presenting to the ED from 1 June 2020 to 17 May 2021. The study compared the PPMC approach (PPMC group) with traditional medical officer-led medication charting approaches in the ED, either an early best-possible medication history (BPMH) group or the usual care group. In the PPMC group, a BPMH was documented promptly soon after arrival in the ED, subsequent to which a collaborative discussion, co-planning, and co-charting of medications were undertaken by both a PPMC-credentialled pharmacist and a medical officer. In the early BPMH group, the BPMH was initially obtained in the ED before proceeding with the traditional approach of medication charting. Conversely, in the usual care group, the BPMH was obtained in the inpatient ward subsequent to the traditional approach of medication charting. Three outcome measures were assessed –the duration from ED presentation to the TCM’s first dose administration (e.g., anti-Parkinson’s drugs, hypoglycaemics and anti-coagulants), the completeness of medication orders, and the conduct of VTE risk assessments. The analysis included 321 TCMs, with 107 per group, and 1048 patients, with 230, 230, and 588 in the PPMC, early BPMH, and usual care groups, respectively. In the PPMC group, the median time from ED presentation to the TCM’s first dose administration was 8.8 h (interquartile range: 6.3 to 16.3), compared to 17.5 h (interquartile range: 7.8 to 22.9) in the early BPMH group and 15.1 h (interquartile range: 8.2 to 21.1) in the usual care group (*p* < 0.001). Additionally, PPMC was associated with a higher proportion of patients having complete medication orders and receiving VTE risk assessments in the ED (both *p* < 0.001). The implementation of the PPMC model not only expedited the administration of TCMs but also improved the completeness of medication orders and the conduct of VTE risk assessments in the ED.

## 1. Introduction

Medication management is an essential component to enhance the quality of patient care and safety in acute care settings. Medication misadventures, such as inappropriate medication use, may be more pronounced in emergency department (ED) settings, where acutely ill patients receive fast-paced care [1,2]. Congestion in the ED is another concern associated with delays in the timely administration of essential treatments [3,4,5]. Other medication safety issues include prescription errors and incomplete medication orders, which are common in patients discharged from the ED [6,7]. Additionally, about a quarter of patients being admitted to a hospital through the ED are not routinely assessed for venous thromboembolism (VTE) prophylaxis on their first day of hospitalisation [8].

Collaborative charting models involving pharmacists have been studied in Australia to improve medication chart accuracy [7,9,10,11]. An example is partnered pharmacist medication charting (PPMC), which has been a promising approach for decreasing medication errors in poly-medicated people admitted to an emergency short-stay unit and general medicine unit [12]. PPMC refers to the charting of initial inpatient medicines by a PPMC-credentialled pharmacist following a clinical conversation with a medical officer who is responsible for patient care [11]. To potentially improve medication charting accuracy and medication safety, the Tasmanian Government’s Department of Health and Tasmanian Health Service (THS) approved a 12-month PPMC pilot project in the Royal Hobart Hospital (RHH) ED, Australia, in May 2020. The hypothesis was that a clinical review by pharmacists trained in PPMC followed by a clinical conversation with ED medical officers prior to chart writing would improve medication-related outcomes in comparison to usual care.

Previous studies have generally focused on the impact of PPMC on medication errors or length of stay [11,13,14]. There is a paucity of evidence on whether PPMC minimises the delay in administering time-critical medicines (TCMs), defined as “medicines where early or delayed administration by more than 30 min from the prescribed time for administration may cause harm to the patient or compromise the therapeutic effect” [15]. The impact of PPMC on the completeness of medication orders and the conduct of VTE risk assessment is also poorly understood. Therefore, this external evaluation aimed to investigate the impact of PPMC on the time from ED presentation to the administration of the first doses of TCMs, completeness of medication orders, and conduct of VTE risk assessment.

## 2. Materials and Methods

### 2.1. Study Design and Setting

This evaluation was a retrospective concurrent, controlled pragmatic trial of adult patients (18 years or older) seeking ED care who had been taking at least one regular medication prior to admission within the period from 1 June 2020 to 17 May 2021. The study included patients subsequently admitted to a general medical unit, emergency medicine unit, or mental health unit. The PPMC project was implemented and evaluated in the Royal Hobart Hospital ED, a 490-bed teaching and referral public hospital. The hospital provides services to approximately a quarter of a million people each year, with over 63,000 annual ED visits [16]. The hospital is Tasmania’s largest hospital and the state’s major referral centre that provides acute, sub-acute, mental health, and aged care inpatient and outpatient services. The specific details regarding the study population, study design, study arms, inclusion and exclusion criteria, and data collection procedures have been comprehensively outlined elsewhere [11].

### 2.2. Study Arms

Three study arms were compared: the PPMC arm, which represented a process ‘redesign’, the early best-possible medication history (BPMH) arm, reflecting a process ‘tweak’, and the usual care arm, denoting a traditional standard of care.

In the PPMC group, a BPMH was promptly documented by an ED pharmacist at the earliest possible time in the ED. The BPMH was obtained through structured patient interviews and secondary sources, such as caregivers, electronic health records, and community pharmacies. Following the clinical review, the pharmacist collaborated with a medical officer to have an interdisciplinary patient-centred discussion and to develop a shared medication treatment plan (SMTP). The co-signed SMTP was placed in the patient’s medical progress notes. Based on the SMTP, medications were then charted by the pharmacist using purple ballpoint ink. The medical officer endorsed each medication order before administration by the nursing staff. A ward pharmacist later conducted a medication reconciliation (MedRec) on the inpatient ward (Figure 1).

The early BPMH group included timely BPMH documentation by an ED pharmacist as early as feasible in the ED. Subsequently, a traditional medication charting approach was followed, wherein a medical officer charted medications in the chart using a black/blue ballpoint pen in the ED. In this approach, no clinical conversations occurred between the PPMC pharmacist and the ED medical officer; the BPMH was available to the medical officer prior to charting. Following this, a ward pharmacist performed a MedRec on the inpatient ward.

The usual care group consisted of patients who underwent the standard admission procedure, which involved the traditional medication charting method. In this process, a medical officer wrote medication charts within the ED using black/blue ballpoint ink. Importantly, there was no pharmacist-collected BPMH or collaborative interdisciplinary conversation between the pharmacist and the medical officer in the ED. A ward pharmacist subsequently conducted a MedRec on the inpatient ward.

### 2.3. Ethics and Site Authorisation Approvals

The University of Tasmania Human Research Ethics Committee approved the study protocol (H0018682). Permission for site authorisation for the research project was also granted from the Tasmanian Department of Health Research Governance Office.

### 2.4. Data Collection

Patients were pragmatically assigned to one of the study arms by ED staff as part of their routine patient care. Eligible study participants from each arm were then randomly selected using an online random number generator: http://izmm.com/random.pl (accessed on 15 July 2021). An independent researcher (TMA) retrospectively collected data from the patients’ medical records and secondary administrative data sets.

### 2.5. Outcome Measures

#### 2.5.1. Time to Time-Critical Medicines

A local THS-specific list of TCMs was used in the study (the drug groups are shown in Figure 2). The duration, in hours, from ED presentation to the TCM’s first dose administration was compared between the three study groups. Only the first dose of a pre-admission TCM that was charted and administered at the hospital was included. TCMs withheld during the admission and those with undocumented indications were excluded.

#### 2.5.2. Completeness of Medication Orders

Medication charting issues relevant to the completeness of medication orders, such as incomplete, unclear, or unsigned orders, were routinely documented by independent ward pharmacists in the Medication Management Action Plan (MedMAP) as part of their regular medication review. An independent researcher (TMA) retrospectively extracted the charting issues from the MedMAP and classified them into incomplete, unclear, or unsigned orders. Only those charting issues specific to the medication orders for the initially charted medicines were included. Examples included the following.

Incomplete order: A dose and/or frequency was omitted in the medication order. An example of a pharmacy note: “Currently charted hydrochlorothiazide 12.5 mg with no frequency. Please add frequency to order”.Unclear order: An order was illegible or edited to the original order. Example: “Quetiapine: currently an illegible order (150 mg? 250 mg). Please re-chart it”.Unsigned order: An order was not signed by a prescriber. Pharmacy note: “Requires doctor’s signature for oxycodone 5 mg (Endone)”.

#### 2.5.3. VTE Risk Assessment

The proportion of patients with a complete VTE risk assessment in their initial National Inpatient Medication Chart (NIMC) was compared between the three groups. Hospital clinicians utilised the THS-specific VTE risk assessment form to screen patients who were being admitted. In the PPMC group, a PPMC pharmacist completed the VTE risk assessment form, discussed it with an ED medical officer and documented the VTE risk assessment and plan, both in the SMTP and NIMC. In the early BPMH and usual care groups, a medical officer completed the VTE risk assessment form and documented the VTE risk assessment and plan in the NIMC. A ward clinical pharmacist subsequently checked patients’ medical records for any documentation of VTE risk assessment at the time of the medication chart review. If a VTE risk assessment had not been conducted, the ward pharmacist brought this issue to the attention of the treating inpatient physician through the MedMAP. Conduct of the VTE risk assessment was then retrospectively extracted from the MedMAP notes by the independent researcher (TMA). During data collection, documented evidence of the VTE risk assessment was further checked in the VTE risk assessment form and SMTP.

### 2.6. Sample Size Calculation

Assuming a beta risk of 0.1 and an alpha risk of 0.05 (two-tailed test), a total of 321 (107 per group) instances of the use of TCMs would be needed to detect a 20% relative decrease (i.e., moderate effect size) in the average time from ED presentation to the first dose administered with PPMC compared to the early BPMH alone or usual care. Samples of TCMs were randomly selected from each TCM group using a proportionate random sampling technique. To account for any potential clustering effect, a subgroup analysis was further conducted. This sub-group analysis included the TCM in a patient who had only one TCM, or one randomly selected TCM in a patient who had two or more TCMs. The completeness of medication orders and VTE risk assessment were secondary outcomes of the medication discrepancies/errors study [11].

### 2.7. Statistical Analysis

Categorical variables were summarised using frequencies and percentages and compared using Pearson’s chi-square test or Fisher’s exact test, as appropriate. We assessed the normality of the continuous variables using both the Shapiro–Wilk test and graphical methods. Ordinal and non-normally distributed continuous data were presented using medians (25% to 75% interquartile range [IQR]) and compared using the Kruskal–Wallis test with Dunn’s post-hoc test. We considered a *p*-value of less than 0.05 as statistically significant for all analyses. Additionally, we adjusted the *p*-values for multiple comparisons using the Benjamini–Hochberg method. All statistical analyses were performed in R^®^ version 4.1.12 (R Foundation for Statistical Computing, Vienna, Austria).

The study was evaluated using SQUIRE 2.0 (Standards for QUality Improvement Reporting Excellence) [17], which serve as a reporting checklist for quality improvement evaluations in health care (Appendix A).

## 3. Results

### 3.1. Patients’ Characteristics

A previously published report [11] presents in-depth the screening and selection of study participants and their demographic and clinical characteristics.

### 3.2. Time to Time-Critical Medicines

Findings of the time to TCMs are summarised in Table 1. A total of 321 TCMs were analysed, with categories of TCMs distributed evenly across the study groups. The time elapsed from ED presentation to the administration of the first TCM dose was significantly different between the study groups (*p* < 0.001). The median times were 8.8 (IQR: 6.3 to 16.3), 17.5 (IQR: 7.8 to 22.9), and 15.1 (IQR: 8.2 to 21.1) hours in the PPMC, early BPMH, and usual care groups, respectively. The median times for the early BPMH group and the usual care group were not significantly different (*p* = 0.22).

After adjusting for the clustering effect (i.e., after including the TCM for a patient with one TCM and one randomly selected TCM for a patient with multiple TCMs), there continued to be significant differences in the median times between the study groups (*p* = 0.007). Patients in the PPMC group had lower median times (9.0 h [IQR: 6.3 to 16.1]) than those in the early BPMH group (16.7 h [IQR: 7.1 to 22.2], *p* = 0.008) and the usual care group (14.2 h [IQR: 7.3 to 21.2], *p* = 0.019).

A TCM group-level analysis generally yielded similar findings, with PPMC shortening the median administration times in the majority of pre-admission TCM categories compared to the early BPMH group or the usual care group (Figure 2). The most substantial reductions in median administration times were observed for insulin, anti-Parkinsonian medications, and corticosteroids and hormones, with subsequent reductions noted for anti-coagulants and oral hypoglycaemic medications.

### 3.3. Completeness of Medication Orders

No medication order for the initially charted medicines in the ED was deemed incomplete or unclear by the reviewing ward pharmacists in the PPMC group; however, one medication order was reported as unsigned. The early BPMH and usual care groups showed a greater number of incomplete, unclear, or unsigned medication orders, with a statistically significant difference in comparison to the PPMC group (*p* < 0.001). The occurrence of incomplete, unclear, or unsigned orders occurred in only one patient (0.4%) in the PPMC group compared to seventeen (7.4%) in the early BPMH group and forty-five (7.7%) in the usual care group, with statistically significant differences between the groups (both *p* < 0.001) (Table 2).

### 3.4. Venous Thromboembolism Risk Assessment

Nearly all (97%, *n* = 223) patients in the PPMC group had their risk for VTE assessed and documented in the ED. Significantly fewer patients in the early BPMH (71.7%, *n* = 165) and usual care group (68.7%, *n* = 404) had their risk for VTE assessed in the ED compared to the PPMC group (both *p* < 0.001).

## 4. Discussion

This study aimed to assess the impact of PPMC on the time to administer the first dose of TCMs, medication order completeness, and VTE risk assessment. The PPMC’s impact was evaluated in comparison with early BPMH alone or usual care. The implementation of the PPMC care model significantly shortened the administration time for pre-admission TCMs compared to early BPMH alone or usual care. The PPMC model also resulted in statistically significant improvements in the completeness of medication orders for initially charted medicines in the ED. A further advantage of the PPMC model was that a higher percentage of patients had their risk for VTE assessed in the ED.

The sub-group analysis, which aimed to adjust for a potential clustering effect, likewise confirmed the significant impact of PPMC on the time to TCMs. While the impact of ED pharmacists on chronic TCMs remains understudied in the existing literature, this finding is consistent with earlier reports of ED-based pharmacist interventions that shortened the administration time of certain acute TCMs in an acute care setting [18,19,20]. These retrospective previous studies highlighted the significant impact of pharmacists specialising in emergency medicine, who improved the initiation of acute stroke medications in the context of ischaemic stroke care and enhanced antiepileptic therapy for patients with status epilepticus within the ED.

Whilst all medications should be administered in a timely manner, the administration of TCMs requires special consideration. Unintentionally delaying their administration could result in harm to the patient (including fatal outcomes), as it could delay symptoms being controlled or worsen the condition [21]. The close collaboration between pharmacists and medical officers enabled streamlined processes, reducing the time lapse between medication orders and administration.

The study did not consider the timing of TCM doses before ED presentation, focusing solely on the interval between ED presentation and the first dose administration at the hospital due to data limitations. To address potential variability in medication timing, a subgroup analysis using TCM groups was conducted (Figure 2), assuming similar administration frequencies within the TCMs group. The subgroup analysis partially addressed this limitation; however, further research is needed to explore the impact of pre-ED arrival medication timings on any delays in the administration of the first dose of TCMs at the hospital.

For charts written using the PPMC model of care, medication orders were more legible and complete than the traditional medical charting approaches. Complete and legible medication charts are vital for communicating the intentions of written medication orders. They facilitate consistent communication of patients’ medication management amongst treating clinicians and departments. Incomplete medication orders may increase the risk of adverse drug events [22]. The findings for the completeness of medication orders were based on the ward clinical pharmacists’ comments, which was a potential limitation. These comments were made in real-time during the routine medication chart review and were communicated to a treating physician via MedMAP. Different ward pharmacists may have varied “thresholds” for documenting MedMAP issues, particularly those requiring subjective judgments, such as illegible handwriting. For future studies, we suggest assessing the safety and completeness of medication orders using prospective methods with the nationally standardised Australian Commission on Safety and Quality in Health Care audit tool [23].

The percentage of patients with a complete VTE risk assessment in the ED was significantly higher in the PPMC group (97%) than in the early BPMH group (72%) and usual care group (69%). This finding was similar to a previous study that reported a higher rate of VTE prophylaxis assessment and charting in the collaborative charting group (98%) than in the control group (65%) [24]. Another previous study reported that no inappropriate orders for VTE prophylaxis were identified for medications charted using the PPMC model of care [9]. VTE is relatively common among hospitalised patients and causes a substantial morbidity and mortality burden [25,26,27]. Based on the epidemiological evidence from Australian reports, VTE accounts for approximately 7% of all hospital deaths and $1.72 billion in annual costs in 2008 [28,29,30]. Implementing new practices, such as PPMC, is required to provide early risk assessments to prevent hospital-acquired VTE.

There are certain limitations of the pragmatic evaluation design to be acknowledged. Given that the study was not designed as a prospective randomised controlled trial, potential selection bias could arise. A retrospective design may not have the capacity to fully control unknown confounding variables and may not precisely ascertain the accuracy, timeliness, and completeness of the collected data. Further investigation should focus on the long-term outcomes of the PPMC model on patient care and medication management, including medication adherence and patient outcomes. Future research should also explore the broader applicability of such collaborative models in various healthcare settings to gauge their impact in ensuring optimal medication management.

## 5. Conclusions

With the PPMC model, pre-admission TCMs were administered more promptly following a patient’s hospital presentation compared to early BPMH alone or usual care. Additionally, PPMC resulted in more complete medication orders and improved VTE risk assessments in the ED. Partnering PPMC pharmacists with medical officers to jointly optimal medication plans that address acute medical issues in the ED is important to promote the quality use of medicines in patients taking at least one regular pre-admission medication. The findings implied that closer interdisciplinary collaboration between pharmacists and medical officers could support shared clinical decision-making and facilitate timely patient care, and thus support the continuation of PPMC in the ED.

## Figures and Tables

**Figure 1 pharmacy-12-00071-f001:**
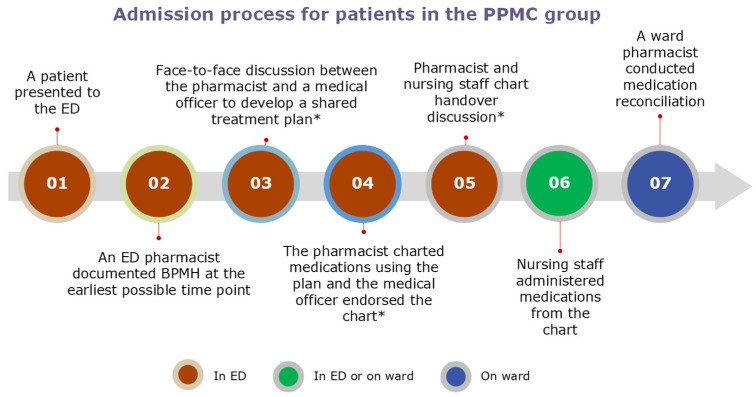
Admission process in the PPMC arm. * The admission process in the early BPMH group did not include steps 3 to 5, whereas the usual care group did not include steps 2 to 5, with the processes instead following a traditional medication charting approach conducted by the medical officer. Abbreviations: BPMH, best-possible medication history; ED, emergency department; PPMC, partnered pharmacist medication charting.

**Figure 2 pharmacy-12-00071-f002:**
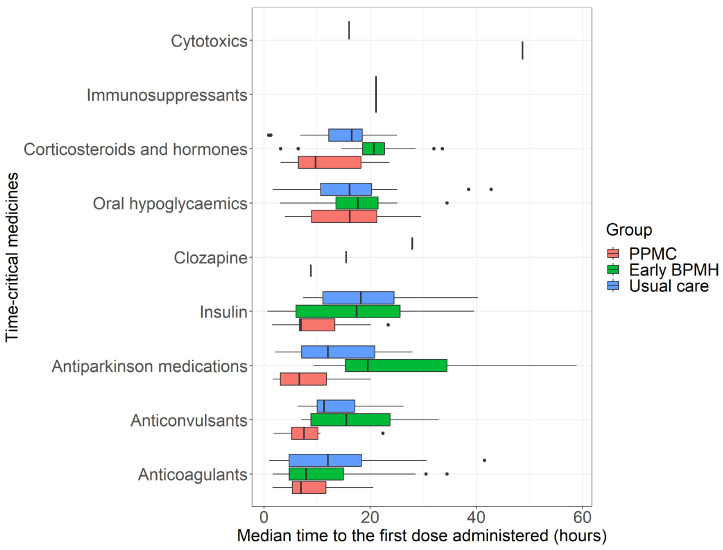
Median times between ED presentation and administration of the first dosage of time-critical medicines. Abbreviations: BPMH, best-possible medication history; ED, emergency department; PPMC, partnered pharmacist medication charting.

**Table 1 pharmacy-12-00071-t001:** Frequency of pre-admission time-critical medicine groups and their overall median times from ED presentation to the first dose administration.

Variables	Before Accounting for a Potential Clustering Effect				After Accounting for a Potential Clustering Effect			
	PPMC (*n* = 107 TCMs)	Early BPMH (*n* = 107 TCMs)	Usual Care (*n* = 107 TCMs)	*p*-Value	PPMC (*n* = 77 TCMs)	Early BPMH (*n* = 76 TCMs)	Usual Care (*n* = 91 TCMs)	*p*-Value
Group, n (%)								
Anti-coagulants	24 (22%)	23 (21%)	24 (22%)	0.98 *	16 (21%)	21 (28%)	22 (24%)	0.61 *
Anti-convulsants	9 (8.4%)	9 (8.4%)	8 (7.5%)	0.96 *	6 (7.8%)	7 (9.2%)	6 (6.6%)	0.82 *
Anti-Parkinson’s medications	9 (8.4%)	9 (8.4%)	10 (9.3%)	0.96 *	7 (9.1%)	7 (9.2%)	8 (8.8%)	0.99 *
Clozapine	1 (0.9%)	1 (0.9%)	1 (0.9%)	>0.99 †	1 (1.3%)	1 (1.3%)	1 (1.1%)	>0.99 †
Corticosteroids and hormones	14 (13%)	19 (18%)	16 (15%)	0.63 *	9 (12%)	13 (17%)	14 (15%)	0.63 *
Cytotoxic drugs	0 (0%)	1 (0.9%)	1 (0.9%)	>0.99 †	-	-	-	-
Immunosuppressants	0 (0%)	1 (0.9%)	0 (0%)	>0.99 †	-	-	-	-
Insulin	17 (16%)	15 (14%)	16 (15%)	0.93 *	15 (19%)	11 (14%)	13 (14%)	0.60 *
Oral hypoglycaemic agents	33 (31%)	29 (27%)	31 (29%)	0.83 *	23 (30%)	16 (21%)	27 (30%)	0.37 *
Median time, hours (IQR)	8.8 (6.3, 16.3)	17.5 (7.8, 22.9)	15.1 (8.2, 21.1)	<0.001 ‡,§	9.0 (6.3, 16.1)	16.7 (7.1, 22.2)	14.2 (7.3, 21.2)	0.007 ‡,¶

Abbreviations: BPMH, best-possible medication history; ED, emergency department; IQR; interquartile range; *n*, number; PPMC, partnered pharmacist medication charting; TCMs, time-critical medicines. * Pearson’s chi-square test. † Fisher’s exact test, ‡ Kruskal–Wallis rank sum test. § Dunn’s post-hoc test: <0.001, PPMC vs. early BPMH; 0.002, PPMC vs. usual care; 0.22, early BPMH vs. usual care. ¶ Dunn’s post-hoc test: 0.008, PPMC vs. early BPMH; 0.019, PPMC vs. usual care; 0.53, early BPMH vs. usual care.

**Table 2 pharmacy-12-00071-t002:** Completeness of medication orders for initially charted medicines in the ED.

Variables, *n* (%)	Study Group			*p*-Value *
	PPMC (*n* = 230 Patients)	Early BPMH (*n* = 230 Patients)	Usual Care (*n* = 588 Patients)	
Total incomplete, unclear, and unsigned orders	1	25 †	62 †	<0.001
Patients with at least one incomplete, unclear, or unsigned order	1 (0.4%)	17 (7.4%)	45 (7.7%)	<0.001
Patients with incomplete orders, *n* (%)				<0.001
No incomplete order	229 (100%)	217 (94%)	552 (94%)	
1 incomplete order	0 (0%)	9 (3.9%)	27 (4.6%)	
2 incomplete orders	0 (0%)	2 (0.9%)	7 (1.2%)	
3 incomplete orders	0 (0%)	2 (0.9%)	2 (0.3%)	
Patients with unclear orders, *n* (%) †				0.20
No unclear order	230 (100%)	227 (99%)	577 (98%)	
1 unclear order	0 (0%)	2 (0.9%)	8 (1.4%)	
2 unclear orders	0 (0%)	0 (0%)	3 (0.5%)	
3 unclear orders	0 (0%)	1 (0.4%)	0 (0%)	
Patients with unsigned orders, *n* (%)				0.41
No unsigned order	229 (99.6%)	229 (99.6%)	587 (99.8%)	
1 unsigned order	1 (0.4%)	1 (0.4%)	1 (0.2%)	

* Fisher’s exact test. † Includes orders that were edited to the original order (*n* = 1 in the early BPMH group and *n* = 1 in the usual care group). Abbreviations: BPMH, best-possible medication history; ED, emergency department; PPMC, partnered pharmacist medication charting.

## Data Availability

The data analysed in this study were obtained from the Tasmanian Government Department of Health and collected from patients’ records. Human research ethics committee and site authorisation approvals are required to access the data.

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
