# Peer review of "Redesigning Medication Management in the Emergency Department: The Impact of Partnered Pharmacist Medication Charting on the Time to Administer Pre-Admission Time-Critical Medicines, Medication Order Completeness, and Venous Thromboembolism Risk Assessment"

_pharmacy, 2024, doi:10.3390/pharmacy12020071_

Round 1
Reviewer 1 Report
Comments and Suggestions for Authors
This manuscript outlines the findings of the study titled "Redesigning Medication Management in the Emergency Department: The Impact of Partnered Pharmacist Medication Charting on Time to Administer Pre-Admission Time-Critical Medicines, Medication Order Completeness, and Venous Thromboembolism Risk Assessment." The topic addressed is undoubtedly relevant, and the clarity of the study's design and procedures is commendable. However, I recommend that the authors articulate more specific aims, given the profound impact medication management can have on patient health outcomes.
While the study exhibits transparency in acknowledging its limitations and attempts to mitigate them through references, there is room for improvement in the level of detail provided in the objectives. Additionally, I have minor suggestions to enhance the manuscript. Notably, Figure 1 requires enhancement as the current font size impedes interpretation, warranting enlargement for improved readability.
Moreover, the conclusion appears overly generalized and could benefit from refinement. By specifying key findings and their implications, the conclusion can effectively encapsulate the significance of the study's outcomes.
Comments on the Quality of English Language
Moderate editing of English language required
Author Response
Reviewer 1
- This manuscript outlines the findings of the study titled "Redesigning Medication Management in the Emergency Department: The Impact of Partnered Pharmacist Medication Charting on Time to Administer Pre-Admission Time-Critical Medicines, Medication Order Completeness, and Venous Thromboembolism Risk Assessment." The topic addressed is undoubtedly relevant, and the clarity of the study's design and procedures is commendable. However, I recommend that the authors articulate more specific aims, given the profound impact medication management can have on patient health outcomes.
Response: We appreciate your positive evaluation. We have taken into account the comments and made the necessary adjustments accordingly.
- While the study exhibits transparency in acknowledging its limitations and attempts to mitigate them through references, there is room for improvement in the level of detail provided in the objectives. Additionally, I have minor suggestions to enhance the manuscript. Notably, Figure 1 requires enhancement as the current font size impedes interpretation, warranting enlargement for improved readability.
Response: Based on the comment, we have improved the figure (now labelled as Figure 2 in the revision). (Line 230]
- Moreover, the conclusion appears overly generalized and could benefit from refinement. By specifying key findings and their implications, the conclusion can effectively encapsulate the significance of the study's outcomes.
Response: Based on the comment, we have reworked the conclusion as follows:
“Partnering PPMC pharmacists with medical officers to jointly develop medication plans that address acute medical issues in the ED is important to promote the quality use of medicines in patients taking at least one regular pre-admission medication. The findings implied that closer interdisciplinary collaboration between pharmacists and medical officers could support shared clinical decision-making and facilitate timely patient care, and thus support the continuation of PPMC in the ED.” [Lines 326–331]
Reviewer 2 Report
Comments and Suggestions for Authors
Abstract
Was it a trial? intervention? if yes, please mention it.
Describe samples, groups and numbers in each.
Body of the article
In the introduction, define time-critical medicines (TCMs).
In the methods section, start this section with the reasearch design you have used.
Also, you need to organise the article using the appropriate reporting checklist found on Equator: EQUATOR Network | Enhancing the QUAlity and Transparency Of Health Research (equator-network.org). Fill it out and attach it as supplementary file.
Even if the methodlogical details have been published elsewhere, this is an independent article and should be accompanied with such details. Therefore, breifly present them here also.
Your text is full of abbreviations that have not been fully described st the first of apprearance in the text, such as BPMH. Check them all and be sure that they have been fully defined as mentioned. Also, all abreviations should be defined and mentioned under each table.
What is missing here is a figure to summarise the intervention used in your study. The text is not enough and is confusing.
The keyword of interdisciplinary is in the title and keyword, but it has not been defined how this study is interdisciplinary.
The random selection of the participants and assignments to groups should be fully described.
Discussion
It should be started with the study aim.
You used some studies to compare their findings with yours, but did not mention their context, details.
Limitations of the study should be described.
Conclusion
You just repeated your findings. Write implications for practice and future research.
Author Response
Reviewer 2
Abstract
- Was it a trial? intervention? if yes, please mention it.
Response: Incorporating the comment, we have included ‘pragmatic trial’ in the revision. [Line 19]
- Describe samples, groups and numbers in each.
Response: As per the comment, we have incorporated them in the revised version noted below:
“The analysis included 321 TCMs, with 107 per group, and 1048 patients, with 230, 230 and 588 in the PPMC, early BPMH and usual care groups, respectively.” [Lines 27–29]
Body of the article
- In the introduction, define time-critical medicines (TCMs).
Response: Following the suggestion, we have included the definition as advised.
“defined as “medicines where early or delayed administration by more than 30 min from the prescribed time for administration may cause harm to the patient or compromise the therapeutic effect” [1].” [Lines 66–68]
- In the methods section, start this section with the research design you have used.
Response: As suggested, we have reworked the section as follows:
“This evaluation was a retrospective concurrent, controlled pragmatic trial of adult patients (18 years or older) seeking ED care, who had been taking at least one regular medication prior to admission, within the period from June 1, 2020, to May 17, 2021.” [Lines 75–77]
- Also, you need to organise the article using the appropriate reporting checklist found on Equator: EQUATOR Network | Enhancing the QUAlity and Transparency Of Health Research (equator-network.org). Fill it out and attach it as supplementary file.
Response: As suggested, we have now included the EQUATOR in our revision.
“The study was evaluated using SQUIRE 2.0 (Standards for QUality Improvement Reporting Excellence) [17], which serve as a reporting checklist for quality improvement evaluations in health care (Supplementary Material).” [Lines 190–192]
- Even if the methodological details have been published elsewhere, this is an independent article and should be accompanied with such details. Therefore, briefly present them here also.
Response: As suggested, we have reworked the section as follows:
“The study included patients subsequently admitted to a general medical unit, emergency medicine unit or mental health unit. The PPMC project was implemented and evaluated in the Royal Hobart Hospital ED, a 490-bed teaching and referral public hospital. The hospital provides services to approximately a quarter-million people each year, with over 63,000 annual ED visits [2]. The hospital is Tasmania’s largest hospital and the state’s major referral centre that provides acute, sub-acute, mental health and aged care inpatient and outpatient services. The specific details regarding the study population, study design, study arms, inclusion and exclusion criteria, and data collection procedures have been comprehensively outlined elsewhere [3].” [Lines 77–86]
“Three study arms were compared: the PPMC arm, which represented a process ‘redesign’, the early best-possible medication history (BPMH) arm, reflecting a process ‘tweak’, and the usual care arm, denoting a traditional standard of care.” [Lines 88–90]
“Patients were pragmatically assigned to one of the study arms by ED staff as part of their routine patient care. Eligible study participants from each arm were then randomly selected using an online random number generator (http://izmm.com/random.pl). An independent researcher (TMA) retrospectively collected data from the patients’ medical records and secondary administrative data sets.” [Lines 126–130]
- Your text is full of abbreviations that have not been fully described at the first of appearance in the text, such as BPMH. Check them all and be sure that they have been fully defined as mentioned. Also, all abbreviations should be defined and mentioned under each table.
Response: As suggested, we defined BPMH [Line 89] and checked other ones in our revision. We defined the abbreviations in Table 2 as well [Lines 249–250].
- What is missing here is a figure to summarise the intervention used in your study. The text is not enough and is confusing.
Response: As suggested, we have now included a figure (Figure 1). [Lines 101–108]
Figure 1. Admission process in the PPMC arm.
*The admission process in the early BPMH group did not include steps 3 to 5, whereas the usual care group did not include steps 2 to 5, with the processes instead following a traditional medication charting approach conducted by the medical officer.
Abbreviations: BPMH, best-possible medication history; ED, emergency department; PPMC, partnered pharmacist medication charting
- The keyword of interdisciplinary is in the title and keyword, but it has not been defined how this study is interdisciplinary.
Response: As suggested, we have now included the definition in the revision.
“Following clinical review, the pharmacist collaborated with a medical officer to have an interdisciplinary patient-centred discussion…” [Lines 94–95]
- The random selection of the participants and assignments to groups should be fully described.
Response: As suggested, we have now addressed both points in the revision as shown below.
“Patients were pragmatically assigned to one of the study arms by ED staff as part of their routine patient care. Eligible study participants from each arm were then randomly selected using an online random number generator (http://izmm.com/random.pl). An independent researcher (TMA) retrospectively collected data from the patients’ medical records and secondary administrative data sets.” [Lines 126–130]
Discussion
- It should be started with the study aim.
Response: Based on the comment, we have now included the aim in the revision.
“This study aimed to assess the impact of PPMC on time to administer the first dose of TCMs, medication order completeness and VTE risk assessment. The PPMC’s impact was evaluated in comparison with early BPMH alone or usual care.” [Lines 257–259]
- You used some studies to compare their findings with yours, but did not mention their context, details.
Response: Based on the comment, we have now included it in the revision.
“These previous retrospective studies highlighted the significant impact of pharmacists specialising in emergency medicine, who improved the initiation of acute stroke medications in the context of ischaemic stroke care and enhanced antiepileptic therapy for patients with status epilepticus within the ED.” [Lines 269–273]
- Limitations of the study should be described.
Response: Based on the comment, we have now included the limitations.
“There are certain limitations of the pragmatic evaluation design to be acknowledged. Given that the study was not designed as a prospective randomised controlled trial, potential selection bias could arise. A retrospective design may not have the capacity to fully control unknown confounding variables and may not precisely ascertain the accuracy, timeliness and completeness of collected data. Further investigation should focus on the long-term outcomes of the PPMC model on patient care and medication management, including medication adherence and patient outcomes. Future research should also explore the broader applicability of such collaborative models in various healthcare settings to gauge its impact in ensuring optimal medication management.” [Lines 313–321]
Conclusion
- You just repeated your findings. Write implications for practice and future research.
Response: As suggested, we have now included the implications in our revision.
“Partnering PPMC pharmacists with medical officers to jointly develop medication plans that address acute medical issues in the ED is important to promote the quality use of medicines in patients taking at least one regular pre-admission medication. The findings implied that closer interdisciplinary collaboration between pharmacists and medical officers could support shared clinical decision-making and facilitate timely patient care, and thus support the continuation of PPMC in the ED.” [Lines 323–331]
References
- Government of Western Australia Department of Health, Guiding Principles for Timely Administration of Medications. . In 2020.
- Department of Health and Human Services Royal Hobart Hospital. Available online: https://www.dhhs.tas.gov.au/hospital/royal-hobart-hospital [cited 2022 September 13].
- Atey, T. M.; Peterson, G. M.; Salahudeen, M. S.; Bereznicki, L. R.; Simpson, T.; Boland, C. M.; Anderson, E.; Burgess, J. R.; Huckerby, E. J.; Tran, V.; Wimmer, B. C., Impact of Partnered Pharmacist Medication Charting (PPMC) on Medication Discrepancies and Errors: A Pragmatic Evaluation of an Emergency Department-Based Process Redesign. Int. J. Environ. Res. Public Health 2023, 20, (2), 1452.
Round 2
Reviewer 2 Report
Comments and Suggestions for Authors
Nothing more.